# Neudesin Neurotrophic Factor Promotes Bovine Preadipocyte Differentiation and Inhibits Myoblast Myogenesis

**DOI:** 10.3390/ani9121109

**Published:** 2019-12-10

**Authors:** Xiaotong Su, Yaning Wang, Anqi Li, Linsen Zan, Hongbao Wang

**Affiliations:** 1College of Animal Science and Technology, Northwest A&F University, Yangling 712100, China; xiaotongsu1114@163.com (X.S.); wangyn1992@outlook.com (Y.W.); 13562170853@163.com (A.L.); zanlinsen@163.com (L.Z.); 2National Beef Cattle Improvement Centre, Yangling 712100, China

**Keywords:** *NENF*, differentiation, bovine, myoblasts, preadipocytes

## Abstract

**Simple Summary:**

Neudesin neurotrophic factor (NENF) is a secreted protein that was significantly inhibited in the fat-muscle co-culture system in our previous study. However, studies on NENF regulation of bovine muscle development and involvement in the cross-talk between adipose tissue and skeletal muscle have not been reported. Hence, the aim of this study was to clarify the roles of *NENF* in bovine myoblast and preadipocyte differentiation. In this study, we first examined the spatial expression patterns of *NENF* in different tissues and found that *NENF* was highly expressed in the muscle of four-day-old and 24-month-old Qinchuan cattle. Compared with 4-day-old Qinchuan cattle, the expression level of *NENF* was significantly up-regulated in 24-month-old bovine adipose tissue. Then, we detected the expression pattern of the *NENF* gene in bovine preadipocyte and myoblast differentiation and found that the expression of *NENF* mRNA peaks at day 6 during preadipocyte differentiation and peaks at day 4 during myoblast differentiation. Furthermore, we found that the endogenous knockdown of *NENF* inhibited the differentiation of preadipocytes and promoted the differentiation of myoblasts. These findings not only lay the foundation for the construction of regulatory pathways during fat and muscle differentiation but also provide a theoretical basis for molecular breeding of beef cattle.

**Abstract:**

Neudesin neurotrophic factor (NENF) is a secreted protein that is essential in multiple biological processes, including neural functions, adipogenesis, and tumorigenesis. In our previous study, NENF was significantly inhibited in the bovine adipocytes-myoblasts co-culture system. However, studies on NENF regulation of bovine muscle development and involvement in the cross-talk between adipose tissue and skeletal muscle have not been reported. Hence, the aim of this study was to clarify the functional roles of *NENF* in bovine preadipocytes and myoblasts. Real-time quantitative PCR (RT-qPCR) was performed to examine the spatial expression patterns of *NENF* in different tissues, and the results showed that *NENF* was highly expressed in the muscle of four-day-old and 24-month-old Qinchuan cattle. Compared with four-day-old Qinchuan cattle, the expression level of *NENF* was significantly up-regulated in 24-month-old bovine adipose tissue. To explore the roles of *NENF* in bovine myoblast and preadipocyte differentiation, small interfering RNA (siRNA) targeting the *NENF* gene were transfected into bovine preadipocytes and myoblasts to knock down the expression of the *NENF* gene. The results showed that the knockdown of *NENF* in differentiating adipocytes attenuated lipid accumulation, decreased the mRNA expression of adipogenic key marker genes *PPARγ, CEBPα, CEBPβ, FASN,* and *SCD1*, and decreased the protein expression of PPARγ, CEBPα, and FASN. The formation of myotubes was significantly accelerated, and the mRNA expression levels of myogenic marker genes *MYOD1, MYF5, MYF6, MEF2A, MEF2C,* and *CKM,* and the protein expression levels of MYOD1, MYF6, MEF2A, and CKM were up-regulated in myoblasts where *NENF* was knocked down. In short, the knockdown of *NENF* inhibited preadipocyte differentiation and promoted myoblast myogenesis.

## 1. Introduction

Beef is one of the most widely consumed meats in the world. High and consistent quality beef products are increasingly welcomed. Studies on the quality of beef have, therefore, continued to increase. Among the many factors affecting the quality of beef, intramuscular fat (IMF) plays an important role, affecting the tenderness, juiciness, and flavor of beef, especially for the production of “snowflake beef” [1,2,3,4]. At the same time, the price of marbled beef is also very high, with significant economic value. Therefore, beef cattle breeders around the world are trying to find ways to increase the IMF deposition in beef.

IMF, as the main form of fat deposition, is the material basis for the formation of marbling in beef. As the IMF content continues to increase, the white adipose tissue interposed between the muscle fibers becomes more and more obvious, forming a red-and-white phase-like marbled beef [5,6,7]. At the cellular level, marbled beef is composed of muscle cells and fat cells. The two cells are heterogeneously distributed and can interact through autocrine and paracrine pathways [8,9,10,11,12,13]. Therefore, identifying the key genes that co-regulate preadipocyte and myoblast differentiation is critical to improving our understanding of the molecular mechanisms that influence bovine fat and muscle growth and development.

NENF, originally annotated in mouse embryos, encodes the secreted protein neudesin that consists of 171 amino acids with neurotrophic activity [14]. NENF is a member of the membrane-associated progesterone receptor (MAPR) protein family, which consists of four members with a typical cytochrome b5-like heme/steroid binding domain [15,16,17]. Members of the MAPR family include progesterone receptor membrane component 1 (PGRMC1), progesterone receptor membrane component 2 (PGRMC2), NENF, and neuferricin, which are abundantly expressed in the central nervous system and exert neurotrophic effects in nerve cells [15]. As an extracellular heme-binding protein, NENF requires heme to bind to its cytochrome b5-like heme/steroid binding domain to exert its neurotrophic activity [18]. *NENF* is abundantly expressed in the brain and spinal cord of mouse embryos, but is also widely distributed in various tissues after birth and is essential in multiple biological processes including neural functions, adipogenesis, and tumorigenesis [19,20,21,22,23,24,25].

Most studies of NENF have focused on neural functions. *NENF* has been shown to be expressed early in cultured neuronal cells and exhibits neurotrophic activity [14]. Many studies have reported that NENF has a positive effect on the proliferation and differentiation of neural precursor cells [19]. NENF has also been identified as the candidate oncogene GIG47, which is expressed in a variety of human cancers and is involved in tumorigenesis [23,24]. Only one study investigated the function of NENF in adipogenesis [22]. Studies on 3T3-L1 murine adipocytes showed that interference with the *NENF* gene significantly inhibited the mitogen-activated protein kinase (MAPK) pathway activation and promoted lipogenesis, suggesting that NENF may be a negative regulator of early adipogenesis [22].

To date, studies on the regulation of muscle growth and development by the *NENF* gene have not been reported. Only one study explored the roles of *NENF* in fat deposition in 3T3-L1 cell lines [22], taking into account differences between species, the function of NENF in fat deposition still needs further exploration. The aim of this study was to clarify the roles of the *NENF* gene in bovine myoblast and preadipocyte differentiation in order to provide a theoretical basis for improving beef meat quality and molecular breeding of beef cattle.

## 2. Materials and Methods 

### 2.1. Animals

All animal procedures are in accordance with the Regulations for the Administration of Affairs Concerning Experimental Animals (Ministry of Science and Technology, China, 2004) and approved by the Institutional Animal Care and Use Committee (College of Animal Science and Technology, Northwest A&F University, China, No.2013-23, 20 April 2013). 

A three-day old healthy Qinchuan beef cattle was used for preadipocyte and myoblast isolation and cell culture in this study. Twelve tissue samples, including heart, spleen, lung, kidney, rumen, reticulum, omasum, abomasum, large intestine, small intestine, muscle, and adipose were collected from the four-day-old and 24-month-old Qinchuan cattle. Cattles were born and raised at the experimental farm of the National Beef Cattle Improvement Center (Yangling, China) and humanely slaughtered in a slaughterhouse. 

### 2.2. Isolation, Culture, and Cell Induction of Bovine Adipocytes and Myoblasts

Isolation, culture, and cell induction of bovine preadipocytes and myoblasts were performed as described by Li et al. and Wang et al. [26,27]. Briefly, the primary preadipocytes and myoblasts were isolated separately from perirenal adipose tissue and hind limb muscle. After isolation, the primary preadipocytes were cultured in the complete growth medium of Dulbecco’s Modified Eagle Medium/F-12 (DMEM/F-12, Gibco, Grand Island, NY, USA) containing 10% fetal bovine serum (FBS, Gibco) and 1% penicillin/streptomycin. The primary myoblasts were cultured in complete growth medium containing DMEM/F-12, 20% FBS (Gibco) and 1% penicillin/streptomycin (Hyclone, Logan, Utah, USA). The preadipocytes and myoblasts were cultured at 37 °C in 95% humidity with 5% CO2 (Thermo Fisher Scientific, Waltham, MA, USA). For the differentiation study, the growth medium was switched to differentiation medium when the bovine preadipocytes and myoblasts reached 70–80% confluency. The differentiation medium for preadipocytes was prepared with complete medium containing isobutylmethylxanthine (IBMX, 0.5 mM) (Sigma, Kawasaki City, Japan), insulin (1 mg/mL) (Sigma, Kawasaki City, Japan), and Dexamethasone (1 mM) (Sigma, Saint Louis, MO, USA). The differentiation medium for myoblasts contains DMEM/F-12, 2% horse serum (HS, Gibco), and 1% penicillin/streptomycin. The cell culture medium was changed every two days.

### 2.3. SiRNA Transfection

All siRNAs against the bovine *NENF* gene were synthesized by RiboBio (Guangzhou, China). Sequences are shown in Table 1. Cells were seeded in six-well plates, and when cells reached 70–80% confluence, siRNAs were transfected into bovine preadipocytes and myoblasts with Lipofectamine^TM^ 3000 (Invitrogen, Carlsbad, CA, USA) at a final transfection concentration of 50 nM. Twenty-four hours after transfection, the OPTI-medium was replaced with differentiation medium to induce preadipocytes and myoblasts differentiation.

### 2.4. Dipyrromethene Boron Difluoride (BODIPY) Staining

BODIPY staining was performed to detect adipocyte differentiation, which is a fluorescence-based approach. Adipocytes were fixed with 4% paraformaldehyde for 30 min at room temperature and then incubated with BODIPY dye (Invitrogen) (working solution 1 mg/mL) for 30 min. After incubation, cells were treated with 4’,6-diamidino-2-phenylindole (DAPI) (Gibco, Grand Island, NY, USA) to stain the nuclei for 10 min. Immunofluorescence images were captured by an Olympus IX71 microscope (OLYMPUS, Dalian, China).

### 2.5. Quantitative Real Time-PCR (qRT-PCR)

Total RNA was isolated from tissues, and cultured cells with Trizol reagent (Takara, Mountain View, CA, USA), and then reverse transcription was performed to synthesize cDNA using the Prime Script RT reagent kit (Takara, Mountain View, CA, USA) according to the manufacturer’s instructions. Subsequently, the cDNA was used for qRT-PCR in triplicate wells by the SYBR Green Real-Time PCR Master Mix (Takara, Mountain View, CA, USA) in 7500 Real-Time PCR System (Applied Biosystems, Foster City, CA, USA). Using β-Actin as an internal control in adipocytes and glyceraldehyde-3-phosphate dehydrogenase (GAPDH) as an internal control in myoblasts to normalize the target gene mRNA levels. The relative gene expression was calculated using the 2^−ΔΔCt^ algorithm method [28]. All primer sequences used for qRT-PCR detection are listed in Table 2.

### 2.6. Western Blot Analysis

Cells were washed three times with PBS, and then the total cellular protein was extracted from adipocytes and myoblasts with a protein extraction kit (Solarbio Company, Beijing, China). Protein concentration was quantified using the bicinchonininc acid (BCA) method (Takara). Protein loading buffer was then added to the cellular protein and denatured in a metal bath for 10 min. Equal amounts (20 μg) of protein sample were electrophoresed on a 12% sodium dodecyl sulfate polyacrylamide gel electrophoresis (SDS-PAGE) gel and then transferred to a polyvinylidene fluoride (PVDF) membrane. The membrane was then blocked with QuickBlock™ Western blocking solution (Beyotime Biotechnology, Shanghai, China) for 15 min and incubated with primary antibody against GAPDH (rabbit anti-GAPDH, 1:10,000 Abcam, Cambridge, UK, NP_001029206.1), β-Actin (rabbit anti-β-Actin, 1:5000, NOVUS, NT, HK, NP_776404.2), NENF (rabbit anti-NENF, 1:500, Thermo Fisher Scientific, Waltham, MA, USA, NP_001069887.1), PPARγ (rabbit anti-PPARγ, 1:1000, Boster, Wuhan, China, NP_851367.1), CEBPα (rabbit anti-CEBPα, 1:1000, Abcam, NT, UK, NP_789741.2), FASN (rabbit anti-FASN, 1:2000, Abcam, NT, UK, NP_777087.1), MEF2A (rabbit anti-MEF2A, 1:1000, Abcam, NT, UK, NP_001077107.1), MYOD1 (rabbit anti-MYOD1, 1:1000, Abcam, NT, UK, NP_001035568.2), MYF6 (mouse anti-MYF6, 1:100, Santa Cruz Biotechnology, Santa Cruz, CA, USA, NP_861527.1), CKM (mouse anti-CKM, 1:200, Santa Cruz Biotechnology, Santa Cruz, CA, USA, NP_777198.2) overnight at 4 °C. After 1 h incubation at room temperature with Immunoglobulin G (IgG)-Goat anti-Rabbit Horseradish Peroxidase (HRP) antibodies (1:2000, NOVUS, NT, HK), and m-IgGκ BP-HRP (1:1000, sc-516102, Santa Cruz Biotechnology, Santa Cruz, CA, USA) secondary antibodies, protein bands were visualized using a Gel Doc™ XR+ Gel Documentation System (Bio-Rad, Hercules, CA, USA).

### 2.7. Statistical Analysis

Statistical analysis was performed using Independent-samples t-test and one-way analysis of variance (ANOVA) in GraphPad Prism 6.0 (GraphPad Software Inc, San Diego, CA, USA). All data were presented as mean ± SD and *p < 0.05*, *p* < 0.01 were considered to be statistically significant and extremely significant, respectively.

## 3. Results

### 3.1. Tissue Expression Profiles of the Bovine NENF Gene

Figure 1a,b show the relative expression levels of the *NENF* gene in the four-day-old and 24-month-old Qinchuan cattle tissues and organs (including heart, spleen, lung, kidney, rumen, reticulum, omasum, abomasum, large intestine, small intestine, muscle, and adipose), respectively. The results showed that in the four-day-old Qinchuan cattle, *NENF* was predominantly expressed in the heart and muscle, followed by the kidney, large intestine, rumen, omasum, abomasum, reticulum, small intestine, spleen, adipose, and lung. In the 24-month-old Qinchuan cattle, significantly high transcript levels were observed in heart and muscle, followed by the large intestine, kidney, adipose, abomasum, omasum, spleen, lung, reticulum, small intestine, and rumen. The *NENF* mRNA level was elevated in the muscle of 24-month-old adult cattle compared with that in the 4-day-old neonatal calf (*p < 0.01*). In addition, compared with the four-day-old Qinchuan cattle, the relative expression level of *NENF* mRNA in adipose tissue in the 24-month-old Qinchuan cattle was significantly up-regulated (Figure 1c). Since 24 months is an important period of fat deposition in cattle, we speculate that the *NENF* gene may have an effect on the development of adipose tissue in cattle.

### 3.2. NENF Gene Expression Patterns during Differentiation of Bovine Adipocytes and Myoblasts

In order to clarify the expression pattern of the *NENF* gene in bovine adipocytes and myoblasts, the mRNA level of *NENF* was detected at 0, 2, 4, 6, 8, and 10 days after the induction of differentiation. As shown in Figure 2a, the mRNA expression level of *NENF* continued to increase until it reached its peak on the sixth day and then declined slightly during adipocyte differentiation. In contrast to the induction of adipocytes, the mRNA level of *NENF* peaked at day 4 after the induction of myoblasts. Then, from the fourth day to the eighth day, the mRNA expression level decreased slightly, and on the 10^th^ day, it increased again (Figure 2b). These results indicate that the the mRNA expression level of *NENF* is constantly changing with the differentiation time.

### 3.3. Detection of the Interference Efficiency of siRNAs against NENF in Adipocytes and Myoblasts

To determine the interference efficiency of the three siRNAs that interfere with *NENF* gene expression, the siRNAs were transfected into bovine preadipocytes and myoblasts. Twenty-four hours after transfection, the RNA was extracted to detect the interference efficiency. The results showed that siRNA-3 had higher interference efficiency than the negative control group, reaching 95% and 89% in adipocytes and myoblasts, respectively (Figure 3a,b). In addition, cell morphology did not change compared to the control group, so siRNA-3 can be used in subsequent experiments. Subsequently, the interference efficiency of siRNA-3 was detected at 0, 2, 4, and 6 days after induction. Compared with the control group, the relative mRNA expression level of *NENF* was significantly down-regulated in adipocytes and myoblasts transfected with si*NENF* (Figure 3c,d). Western blot analysis showed that NENF protein expression also significantly decreased in si*NENF* transfected adipocytes and myoblasts on day 4. These results indicated that interference with the *NENF* gene by siRNA transfection in adipocytes and myoblasts were effective and successful.

### 3.4. NENF Knockdown Inhibits Adipocyte Differentiation

To investigate the effect of *NENF* in regulating adipocyte differentiation, siRNA was transfected into primary bovine preadipocytes. BODIPY staining results indicated the bovine adipocytes treated with si*NENF* displayed weaker fluorescence intensity and smaller lipid droplets at day 4 and day 6, compared with the control group (Figure 4a,b). Based on the roles of *NENF* in preadipocyte differentiation, the relative expression levels of adipocyte differentiation marker genes *PPARγ*, *CEBPα*, and *CEBPβ* and lipid metabolism key genes *FABP4*, *FASN*, and *SCD1*, were also examined using RT-qPCR. The results showed that the mRNA expression levels of *PPARγ*, *CEBPα*, *CEBPβ*, *FASN*, and *SCD1* were significantly suppressed in *NENF* knockdown adipocytes (Figure 4c–h). The protein expression levels of PPARγ, CEBPα, and FASN were also decreased (Figure 4i), consistent with mRNA expression levels.

### 3.5. NENF Knockdown Promotes Myoblast Differentiation

To determine whether knockdown of *NENF* gene expression could affect myoblast differentiation into myotubes, a *NENF* interference assay was performed using primary myoblasts. The changes in myoblast morphology were analyzed after *NENF* was knocked down, and the differentiation of myoblasts was induced. si*NENF*-transfected myoblasts formed much more and longer myotubes than the negative control (NC) group at day 4 and day 6 (Figure 5a). Moreover, the relative mRNA expression levels of the myoblast differentiation marker genes *MYOD1*, *MYF5*, *MYF6*, *MYOG*, *MEF2A*, *MEF2C*, and *CKM* were also detected at 0, 2, 4, and 6 days after induction, of which *MYF5*, *MYF6*, *MEF2A*, *MEF2C*, and *CKM* were significantly up-regulated in si*NENF*-transfected myoblasts (Figure 5b–h). Western blot analysis showed that the protein expression of MYOD1, MYF6, MEF2A, and CKM were also enhanced (Figure 5i).

## 4. Discussion

The content of IMF, one of the main types of fat in beef, is an important factor in determining the quality of beef. In recent years, with further analysis of the structure of bovine genomes, some functional genes related to bovine fat and muscle growth and development have been gradually discovered. It is vital to identify key genes that co-regulate adipogenesis and myogenesis for fat and muscle development and improving beef quality. 

In another study of this project, high-performance liquid chromatography-mass spectrometry (HPLC/MS) to identify the differentially expressed secreted proteins between co-cultured adipocytes-myoblasts and adipocytes or myoblasts alone, we found that NENF was not detected in the co-culture system. The possible reason may be that the co-culture system inhibited the expression of the *NENF* gene. Based on this previous research, we hypothesized that NENF might play roles both in the regulation of adipogenesis and myogenesis. 

NENF, originally identified as a neurotrophic factor with neurotrophic activity, is abundantly expressed in the brain and spinal cord of mouse embryonic stages [14,19]. *NENF* is highly expressed in white adipose tissue (WAT) of mice, and its expression level is higher than that in the brain, but it is not expressed in brown adipose tissue (BAT) [22]. In the present study, a significant increase in bovine *NENF* gene expression was observed in adipose tissue grown from four days to 24 months, indicating its role in the functional regulation of adipocytes. A previous study demonstrated that the expression of *NENF* decreased in the early stage of differentiation in 3T3-L1 preadipocytes, and increased in the late stage of differentiation, which was consistent with the expression of the adipose differentiation marker genes *ap2, CEBPα,* and *PPARγ,* but not *CEBPβ* [22]. In this study, the expression level of *NENF* in adipocytes showed a trend of increasing first and then decreasing, reaching a peak on day 6. Combined with these findings, the *NENF* gene clearly has different expression patterns among different species, so it is valuable to explore the expression pattern of *NENF* among other species.

Previous studies have shown that NENF significantly promotes neural cell proliferation and neuronal differentiation in neural development [19]. In addition, ectopic expression of *NENF* in MCF7 cells promotes invasiveness and tumorigenesis of breast cancer cells [23]. To explore whether *NENF* has a certain regulatory effect on the differentiation of preadipocytes, we transfected the siRNA targeting *NENF* gene into bovine preadipocytes by RNA interference. A BODIPY staining assay was performed to identify the lipid deposition of bovine adipocytes, and the result showed that knockdown of *NENF* attenuated lipid accumulation. Moreover, several key adipogenic marker genes, including *PPARγ, CEBPα, CEBPβ, FABP4, FASN,* and *SCD1*, were examined to elucidate the mechanism of NENF in preadipocyte differentiation further. We found that interference with NENF reduced the mRNA expression of *PPARγ, CEBPα, CEBPβ, FASN,* and *SCD1* and the protein expression of PPARγ, CEBPα, and FASN, but not FABP4.

The combined results above suggest that NENF has a positive regulatory effect on the differentiation of adipocytes. However, Studies on 3T3-L1 murine adipocytes found that the suppression of endogenous *NENF* gene expression significantly promoted adipogenesis through the MAPK cascade pathway [22]. It has been previously reported that farm animals may exhibit lipid metabolism patterns that are different from rodents [29]. For example, overexpression of the *ZBTB16* gene significantly inhibited the differentiation of 3T3-L1 preadipocytes [30], whereas contradictory results were found during bovine preadipocyte differentiation [31]. It might be possible that differences in fat cells between different species contribute to inconsistent results in adipogenesis.

This is the first report on the regulation of muscle growth and development in the bovine *NENF* gene. In the present study, bovine *NENF* was found to be highly expressed in the muscle of four-day-old and 24-month-old Qinchuan cattle, and the mRNA level was significantly higher in the muscle of 24-month-old adult cattle than that in the four-day-old neonatal calf. Since the postnatal myogenesis in the 4-day-old neonatal calf is more activated than 24-month-old cattle, we speculated that NENF might play negative roles during postnatal myogenesis. Simultaneously, we examined the temporal expression levels of *NENF* at different time points of bovine myoblast development, and the results showed that the expression level of *NENF* in bovine myoblasts increased first and then decreased, reaching the highest level on day 4. In addition, we investigated the effect of NENF in myoblast differentiation using a loss-of-function assay. It has been demonstrated that the knockdown of *NENF* significantly promoted the formation of myotubes. We found that the mRNA expression levels of the myogenic marker genes *MYOD1, MYF5, MYF6, MEF2A, MEF2C,* and *CKM* and the protein expression levels of MYOD1, MYF6, MEF2A, and CKM were up-regulated, but had no effect on the expression of MYOG. Taken together, the endogenous knockdown of *NENF* expression significantly promoted myoblast differentiation. 

The present study provides the first evidence that NENF is a negative regulator of myogenesis in bovine. The Knockdown of *NENF* remarkably accelerates myoblasts differentiation and myotube formation. Since the research about the roles of the *NENF* gene on adipogenesis and myogenesis is limited, more functional studies of NENF in livestock are needed to explore NENF functions on preadipocyte and myoblast differentiation and elucidate the molecular regulation mechanism of NENF during fat and skeletal muscle growth and development.

## 5. Conclusions

Our results showed that NENF is a positive regulator in the differentiation of bovine preadipocytes and a negative regulator in the differentiation of bovine skeletal muscle myoblasts. These findings not only lay a foundation for the construction of regulatory pathways during fat and muscle differentiation but also provide a theoretical basis for molecular breeding of beef cattle.

## Figures and Tables

**Figure 1 animals-09-01109-f001:**
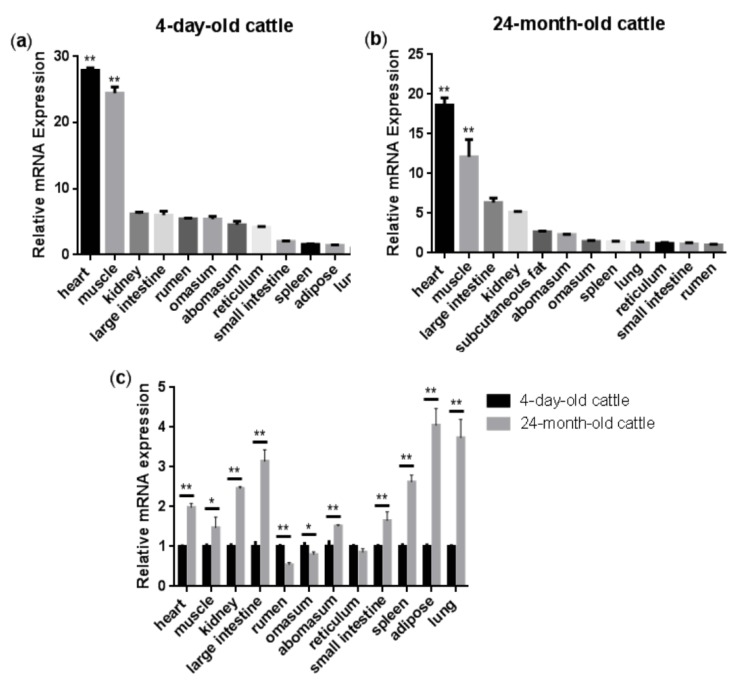
Tissue expression profiles of bovine *NENF* gene: (**a****–b**) Relative mRNA expression levels of *NENF* in the four-day-old and 24-month-old Qinchuan cattle tissues and organs. The expression level of *NENF* is relative to the housekeeping gene β-Actin. When comparing *NENF* expression levels (fold change) among all the tested tissues, we chose lung as the control group for the four-day-old cattle (**a**) and rumen for the 24-month-old cattle (**b**). (**c**) Comparison of relative mRNA expression levels of *NENF* in the four-day-old and 24-month-old Qinchuan cattle. The result was normalized with the housekeeping gene *β-actin* and relative to gene expression in the four-day-old cattle group. Error bars represent s.d. * represent *p <* 0.05 and ** represent *p <* 0.01.

**Figure 2 animals-09-01109-f002:**
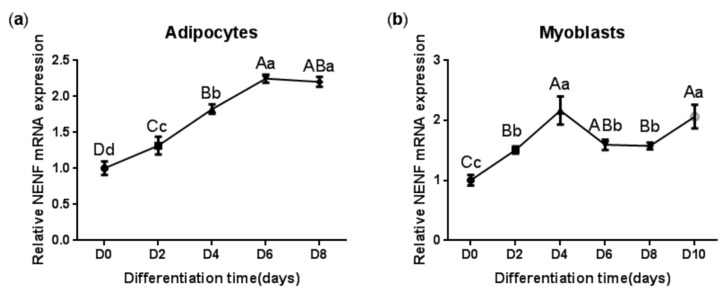
The *NENF* gene expression patterns during differentiation of bovine adipocytes and myoblasts: (**a**) Relative mRNA expression of *NENF* in bovine adipocytes; (**b**) Relative mRNA expression of *NENF* in bovine myoblasts. Error bars represent s.d. Different lowercases among different columns represent *p <* 0.05. Different uppercases among different columns represent *p <* 0.01.

**Figure 3 animals-09-01109-f003:**
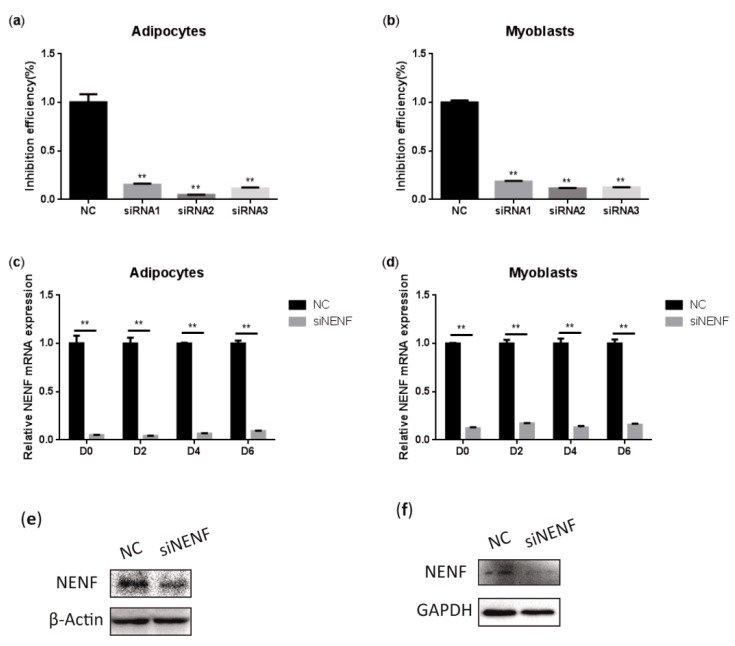
Detection of the interference efficiency of siRNAs against the *NENF* gene in bovine adipocytes and myoblasts: (**a–b**) Detection of the interference efficiencies of three siRNAs after transfection for 24 h in bovine adipocytes (**a**) and myoblasts (**b**). (**c–d**) Detection of the interference efficiency on different days (0, 2, 4, 6 days) of siRNA-3 in bovine adipocytes (**c**) and myoblasts (**d**) at the mRNA expression levels. (**e–f**) Detection of the interference efficiency of siRNA3 at six days (D6) in bovine adipocytes (**e**) and myoblasts (**f**) at the protein expression levels. Error bars represent s.d. * *p* < 0.05; ** *p* < 0.01.

**Figure 4 animals-09-01109-f004:**
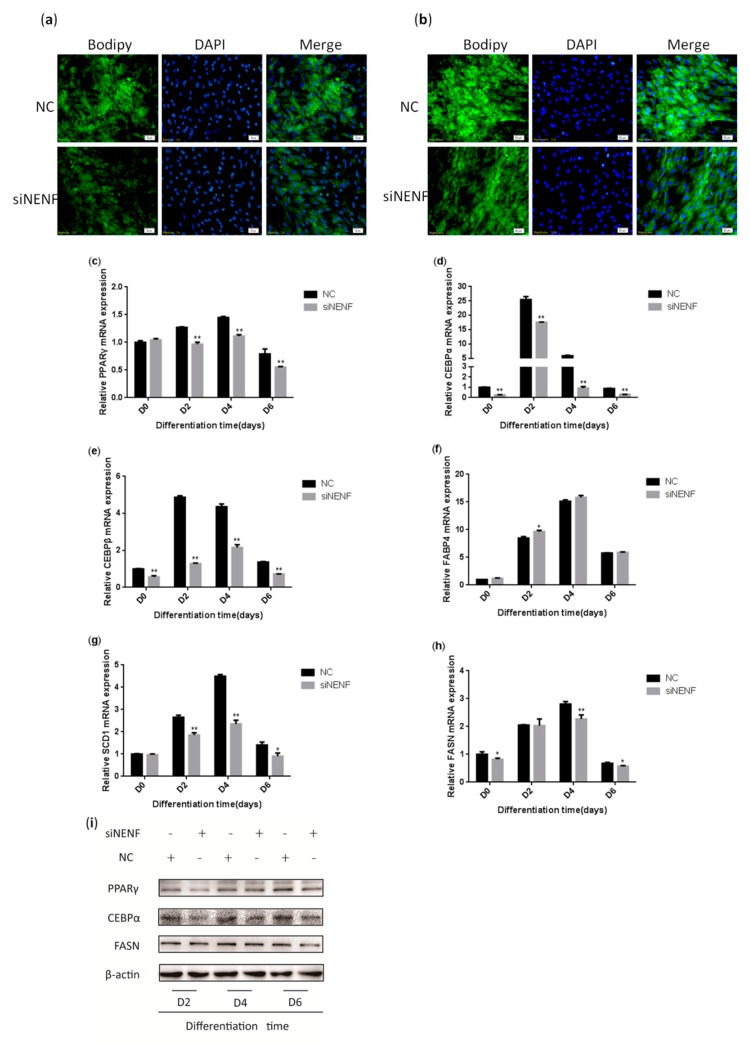
*NENF* knockdown inhibits adipocyte differentiation: (**a–b**) BODIPY (green) and DAPI (blue) staining of bovine adipocytes after being transfected with negative control (NC) and si*NENF* at four days (D4) and six days (D6); (**c–h**) Relative mRNA expression of key adipogenic genes: *PPARγ, CEBPα, CEBPβ, FABP4, SCD1*, and *FASN*; (**i**) Western blot results of PPARγ, CEBPα, and FASN protein expression levels after *NENF* knockdown at two days (D2), four days (D4), and six days (D6).

**Figure 5 animals-09-01109-f005:**
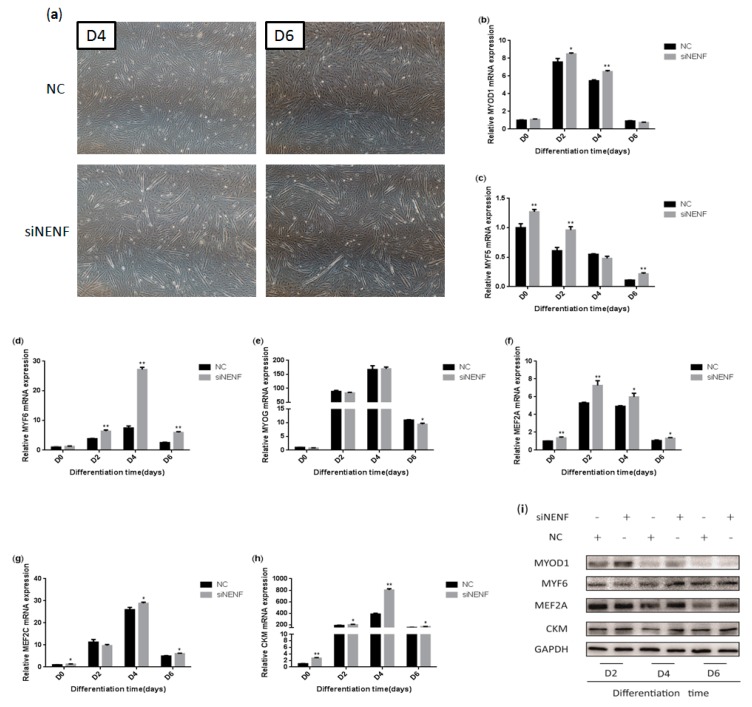
*NENF* knockdown promotes myoblasts differentiation: (**a**) Morphological changes of differentiating bovine skeletal primary myoblasts at four days (D4) and six days (D6) after transfection with si*NENF* and negative control (NC) (OLYMPUS IX71 40×); (**b–h**) Relative mRNA expression of key myogenic genes: *MYOD1, MYF5, MYF6, MYOG, MEF2A, MEF2C,* and *CKM*; (**i**) Western blot results of MYOD1, MYF6, MEF2A, and CKM protein expression levels after *NENF* knockdown at two days (D2), four days (D4), and six days (D6).

**Table 1 animals-09-01109-t001:** Target sequence of siRNA for bovine *NENF*.

Name	Target Sequence (5’–3’)
siRNA-1	CCCGAAGAATCCTCAATGA
siRNA-2	GAAGATCAGCCCATCTACA
siRNA-3	TTGACATAAAGGACGAGTT

Neudesin neurotrophic factor (*NENF*).

**Table 2 animals-09-01109-t002:** Summary information of the genes used for the qRT-PCR in this study.

Gene Name	Accession Numbers	Primer Sequence (5′–3′)	Fragments Size (bp)
*NENF*	NM_001076419	Forward: CATCAGGAGTTTGGCCCTGG	121
		Reverse: AGGTGAGTCGAGTGCTCTGT	
*β-Actin*	NM_173979	Forward: TCTAGGCGGACTGTTAGC	82
		Reverse: CCATGCCAATCTCATCTCG	
*GAPDH*	NM_001034034	Forward: AGTTCAACGGCACAGTCAAGG	124
		Reverse: ACCACATACTCAGCACCAGCA	
*PPARγ*	NM_181024	Forward: TGAAGAGCCTTCCAACTCCC	117
		Reverse: GTCCTCCGGAAGAAACCCTTG	
*CEBPα*	NM_176784	Forward: ATCTGCGAACACGAGACG	73
		Reverse: CCAGGAACTCGTCGTTGAA	
*CEBPβ*	NM_176788	Forward: TTCCTCTCCGACCTCTTCTC	79
		Reverse: CCAGACTCACGTAGCCGTACT	
*SCD1*	NM_173959	Forward: TCCGACCTAAGAGCCGAGAA	200
		Reverse: TGGGCAGCACTATTCACCAG	
*FASN*	NM_001012669	Forward: GGCAAACGGAAAAACGGTGA	183
		Reverse: CTTGGTATTCCGGGTCCGAG	
*FABP4*	NM_174314	Forward: TGAGATTTCCTTCAAATTGGG	101
		Reverse: CTTGTACCAGAGCACCTTCATC	
*MYOD1*	NM_001040478	Forward: AACCCCAACCCGATTTACC	196
		Reverse: CACAACAGTTCCTTCGCCTCT	
*MYF5*	NM_174116	Forward: CCTCTAGTTCCAGGCTCATCTA	90
		Reverse: ACCTCCTTCCTCCTGTGTAATA	
*MYF6*	NM_181811	Forward: GTGATAACTGCCAAGGAAGGAG	93
		Reverse: CGAGGAAATGCTGTCCACGA	
*MYOG*	NM_001111325	Forward: GGCGTGTAAGGTGTGTAAG	85
		Reverse: CTTCTTGAGTCTGCGCTTCT	
*MEF2A*	NM_001083638	Forward: AATGAACCTCACGAAAGCAGAAC	106
		Reverse: TTAGCACATAGGAAGTATCAGGGTC	
*MEF2C*	NM_001046113	Forward: CCTGATGCAGACGATTCAGTAG	123
		Reverse: AAAGTTGGGAGGTGGAACAG	
*CKM*	NM_174773	Forward: GTGGCTGGTGATGAGGAGTC	270
		Reverse: TTTCCCCTTGAACTCACCCG	

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
