# Peer review of "Neudesin Neurotrophic Factor Promotes Bovine Preadipocyte Differentiation and Inhibits Myoblast Myogenesis"

_animals, 2019, doi:10.3390/ani9121109_

Round 1

Reviewer 1 Report

This reviewer think that the authors might not understand the point what I asked.

The authors found NENF mRNA level was elevated in the muscle of 24-month-old adult cattle compared with that in the 4-day-old neonatal calf. From this result, the authors concluded that NENF is highly correlated with muscle development.

In the muscle of 4-day-old neonatal calf, there are still activated muscle stem cells (satellite cells) which contribute to the postnatal muscle growth (postnatal myogenesis or postnatal muscle development). However, in the muscle of 24-month-old cattle, all satellite cells become quiescent and muscle development processes including muscle growth and fiber type transition are completed. If NENF play roles only during muscle development, I assume the NENF expression level is higher in the 4-day-old cattle than that in 24-month-old adult cattle.

From this point of view, in addition to the role in muscle development, NENF may have other roles in adult muscle since NENF is highly expressed in adult skeletal and heart muscles compare with other organs. The authors should discuss about this point in the manuscript.

Author Response

Thanks for the reviewer’s comments. We have already revised this part in the results (from line 218 to 220) and discussions (from line 379 to 382) of the revised manuscript. In the present study, bovine NENF was found to be highly expressed in the muscle of 4-day-old and 24-month-old Qinchuan cattle, and the mRNA level was significantly higher in the muscle of 24-month-old adult cattle than that in the 4-day-old neonatal calf. Since the postnatal myogenesis in the 4-day-old neonatal calf is more activated than 24-month-old cattle, we speculated that NENF may play negative roles during the postnatal myogenesis. Our conclusions from in vitro cell culture experiments also confirm this.

Reviewer 2 Report

The authors have satisfactorily responded to most of the criticisms raised and made the necessary changes to the manuscript.

Author Response

Thanks very much for your comments and suggestions. We are grateful for the detailed comments and suggestions provided by the reviewer, and we believe that your inputs have greatly improved our manuscript.

This manuscript is a resubmission of an earlier submission. The following is a list of the peer review reports and author responses from that submission.

Round 1

Reviewer 1 Report

In this manuscript, Su and colleagues explored the function of NENF in bovine preadipocytes and myoblasts. By using siNENF treatment, the authors showed NENF promotes preadipocyte differentiation. On the other hand, NENF inhibits myoblast differentiation.

Overall, this reviewer think this manuscript is interesting and well organized. The data are presented clearly and experiments seem to be performed rigorously.

However, I have a concern about the following point.

In Figure1, the relative expression of NENF in the muscle in 24-month-old cattle was higher compared with that in 4-day-old cattle. From this result, the authors mentioned that NENF is highly correlated with muscle development. If it’s the case, NENF expression may be probably higher in 4-day-old cattle because myogenesis is completely finished in the muscle of 24-month-old cattle.

To clarify this point, the authors should perform qRT-PCR for differentiation markers of myoblast and adipocyte with 4-day-old and 24-month-old cattle tissues. And if it’s possible, regenerating muscle will be a good positive control to show the involvement of NENF during myogenesis.

Reviewer 2 Report

In this manuscript, the authors evaluate the expression of NENF during bovine muscle and adipose tissue development. Moreover, using primary bovine myoblast and myotubes and gene silencing, they assessed the effects of NENF knockdown on cell differentiation. The topic of this article is interesting, also in light of the growing interest in the crosstalk between adipose tissue and skeletal muscle.

Here are some concerns that should be addressed.

ABSTRACT

The abstract is not very clear and the description of NENF mRNA expression pattern in the different tissues is not described. Thus, the abstract should be revised.

INTRODUCTION

-lane 65. Please provide a brief description of MARP family functions before introducing NENF activities.

-lane 71: “…exhibits certain neurological activity [14].” The term “certain” is too general; please provide more information.

-lane 78-83:  Unpublished observations and personal communications should not be used as references in the introduction section to support the main hypothesis of the present study. My suggestion is to move this sentence into the discussion and replace it with a brief description of what is known about the molecular mechanisms underlying IMF deposition in skeletal muscle. Finally, the reason why the authors hypothesize that NENF might be implicated in this process must be clearly indicated.

-lane 85 “Only one study explored the roles of NENF in fat deposition in 3T3-L1 cell lines”. Please provide the reference at the end of this sentence.

Materials and Methods

lane 91: “Anmials” please correct in “Animals”

lane 117. Section 2.3 “SiRNA transfection” Please provide more information on the transfection agent used.

Lane 135. Did you use the geometric mean between β-Actin and GAPDH genes? The choice of correct(s) reference genes should be carefully verified. Please comment.

Lane 161. Please provide the post-hoc test used after the one-way ANOVA. More details on the statistical methods used should be provided (see later)

RESULTS

Lane 165 “Tissue expression profiles of the bovine NENF gene”. The presentation of the NENF gene expression in the different tissues is not clear to me and should be revised.

   1- Fig 1a shows the relative mRNA expression of NENF in the different tissues. Why did you normalize the NENF expression to the lung for the 4-day-old cattle and to the rumen in 24-month-old cattle? My suggestion is to normalize the NENF gene expression only versus the housekeeping gene(s). This would simultaneously allow describing which tissue express more NENF (i.e. heart and muscle) and if the expression differed between 4-day-old and 24-month-old Qinchuan cattle in some tissues.

   2- In my opinion, the figure 1b is unnecessary since this result might be described more clearly using only fig 1a. However, if you decide to maintain the figure 1b, please make uniform the order of tissues showed in the x-axis of the figure 1a and 1b.

   3-Which is the statistical method used to support the results showed in figure 1a and 1b? In the methods section, you mention only one-way ANOVA but, in this case, you should use two-way ANOVA since you try to find differences not only between tissues but also between 4-day-old and 24-month-old Qinchuan cattle. Please better explain this point.

   4- It seems to me that there are some inconsistencies between the results shown in figure 1a and figure 1b. For example, in the figure 1a the heart tissue of 24-month-old expressed less NENF mRNA compared to 4-day-old cattle. However, the exact opposite is shown in Figure 1b. Please carefully checked the two figures since for most tissues there seems to be this incongruity.

   5- lane 171-172. “In addition, compared with the 4-day-old Qinchuan cattle, the relative expression level of adipose tissue in the 24-month-old Qinchuan cattle was significantly up-regulated”. You probably meant to say ….the relative expression of NENF mRNA in adipose tissue…..

   6-lane 173. “Therefore, it could be preliminarily inferred that NENF is closely related to adipogenesis.” This sentence is not supported by the results obtained since NENF mRNA is up-regulated not only in adipose tissue but also in other several tissues (e.g. heart, spleen, lung ecc..). Please better explain this point.

Lane 184. “…at 0, 2, 4, 6, 8, and 10 days after induction”. Please insert “of differentiation” at the end of this sentence. Moreover, only myoblast differentiation time arrived at 10 days while the adipocytes differentiation time stops at 8 days. Please better describe this paragraph.

Figure 3. The picture of NENF western blot is of very poor quality. Is it possible to replace it with a better one?

DISCUSSION

Lanes 268-270. The sentence “…NENF gene clearly has different expression patterns among different species” is not supported by the results found in the present study. Please better clarify this point.

Lane 304 The sentence “ The present study provides the first evidence that NENF is a novel regulator of myogenesis in bovine, which accelerates differentiation and promotes the formation of myotubes” is not supported by the results found in the present study since NENF know-down promote myogenesis.

It is nor clear to me why NENF expression is higher in muscle tissues (Figure 1) and increased after myoblast differentiation (Figure 2b) but simultaneously seems to inhibit myogenesis (Figure 5). Please better clarify this point.

Did you try to overexpress NENF in preadipocytes and/or myoblasts? What would be the phenotype you would expect?

Did you try to compare the NENF mRNA expression between meat with different IMF contents? This experiment would help to clarify the potential role of NENF during myoblast and preadipocyte differentiation. This limit should be discussed.

Please explain main limitations of this study, as well as of similar in vitro studies and suggest if and how the results may be translated in vivo.

The script would benefit from a major review in type of the manuscript.
